# Computational Study of Novel Semiconducting Sc_2_CT_2_ (T = F, Cl, Br) MXenes for Visible-Light Photocatalytic Water Splitting

**DOI:** 10.3390/ma14164739

**Published:** 2021-08-22

**Authors:** Shaoying Guo, Hao Lin, Jiapeng Hu, Zhongliang Su, Yinggan Zhang

**Affiliations:** 1Fujian Provincial Key Laboratory of Eco-Industrial Green Technology, College of Ecology and Resource Engineering, Wuyi University, Wuyishan 354300, China; linhaosg@wuyiu.edu.cn (H.L.); wqhjp@wuyiu.edu.cn (J.H.); 2Fujian Provincial Key Laboratory of Pollution Control & Resource Reuse, Fujian Normal University, Fuzhou 350007, China; 3Xiamen Products Quality Supervision & Inspection Institute, Xiamen 361005, China; suzhongliang@xmzjy.org; 4College of Materials, Fujian Provincial Key Laboratory of Theoretical and Computational Chemistry, Xiamen University, Xiamen 361005, China

**Keywords:** MXenes, photocatalyst, first-principles calculations

## Abstract

Seeking candidate photocatalysts for photocatalytic water splitting, via visible light, is of great interest and importance. In this study, we have comprehensively explored the crystal structures, electronic properties, and optical absorbance of two-dimensional (2D) Sc_2_CT_2_ (T = F, Cl, Br) MXenes and their corresponding photocatalytic water splitting, under the visible-light region, by first-principles calculations. Herein, we have proposed that 2D Sc_2_CT_2_ MXenes can be fabricated from their layered bulk compounds, alternatively to the traditional chemical etching method. Creatively, we proposed Sc_2_CT_2_ (T = F, Br) as new materials; the band edge alignments of Sc_2_CF_2_ can be tuned to meet the water redox potentials at pH = 8.0. It is highlighted that Sc_2_CF_2_ shows outstanding optical spectra harvested under visible-light wavelength regions, and efficient separation of photo-induced electrons and holes in different zones. These present results provide eloquent evidence and open a new door on the photocatalysis applications of such novel semiconducting MXenes.

## 1. Introduction

Hydrogen production, through photocatalytic water splitting via visible-light irradiation, is a promising solution to solve the environmental issues by achieving low-cost, nonpolluting, and environmentally friendly energy [1,2]. A suitable photocatalyst must fulfill at least the following three requirements: moderate band gaps, appropriate band alignments, and efficient separation of photo-generated electron–hole pairs. This is to say, the band gap of the semiconductor photocatalyst must straddle the redox potential of water, as follows: the conduction band minimum (CBM) must be higher than −4.44 eV, which is the redox potential of H^+^/H_2_; and the valence band maximum (VBM) must be lower than −5.67 eV, which is the redox potential of H_2_O/O_2_ [3,4,5]. Accordingly, 1.23 eV is the smallest band gap for a semiconductor photocatalyst. To exploit solar radiation more efficiently, a value of around 1.8 eV is an ideal gap for an efficient photocatalyst [2,6,7]. Moreover, the efficiency of a photocatalyst must always be hampered by the recombination of photo-induced electron–hole pairs. Therefore, the efficient carrier separation is another vital point in semiconductor photocatalysts. However, finding an efficient semiconductor photocatalyst for hydrogen generation, to meet all the above requirements, is still an ongoing challenge [5].

Recently, two-dimensional (2D) transition metal carbides/nitrides (MXene) were produced from their corresponding MAX phases, where M stands for early transition metal, A stands for IIIA or IVA element, and X represents C or N [8,9,10,11,12,13,14,15,16]. MXenes have been extensively explored for miscellaneous applications as photocatalysts [17,18,19], electromagnetic interference shielding [20], battery anodes [21,22], heavy-metal removal [23], etc. Especially, for the photocatalytic community, monolayer Zr_2_CO_2_ and Hf_2_CO_2_ MXenes have been evaluated as semiconductor photocatalysts. They not only exhibit suitable band gaps and band alignments, but can also separate photo-generated electrons and holes effectively [17,18,19]. Except for Zr_2_CO_2_ and Hf_2_CO_2_, the semiconducting feature was characterized in some other Sc-based MXenes, such as Sc_2_CO_2_ and Sc_2_C(OH)_2_, which possess the band gaps of 2.96 eV and 0.77 eV, respectively [24,25]. Experimentally, a layered Sc_2_CCl_2_ bulk crystal, with a unit cell that is isostructural to a 2D MXene, was successfully synthesized, which implied the feasibility for the fabrication of monolayer Sc_2_CCl_2_ [26]. Theoretically, the calculated optical bandgap of one-layer (1 L), 2 L and 3 L Sc_2_CCl_2_ is about 1.5 eV by the GW + RPA method [27]. Accordingly, the monolayer Sc_2_CCl_2_ is a promising optoelectronic material, due to the excellent light absorbance [27]. In addition, the analogue bulk crystal Y_2_CF_2_ has been directly synthesized through a high-temperature solid-state reaction a while ago [28,29]. The crystal has the ability to photo-reduce protons into H_2_ in the range of visible light [28]. Inspired by these studies, and considering the analogous of F, Cl, and Br, we wonder if Sc_2_CT_2_ MXenes (T = F, Cl, Br) are endowed with outstanding features as efficient semiconductor photocatalysts for water splitting.

Thanks to the progress of the density functional theory, the computational method is now a very important approach for designing new functional material. In the present work, we have comprehensively explored the exfoliate energy, geometrical structure, kinetic and thermal stabilities, as well as electronic properties and optical absorbance of 2D Sc_2_CT_2_ (T = F, Cl, Br) MXenes, using first-principles calculations. Especially, we have extensively explored the possibility of Sc_2_CT_2_ (T = F, Cl, Br) MXenes to be used for visible-light photocatalytic water splitting. Our study can make contributions to the discovery and applications of Sc-based MXenes, as semiconductor photocatalysts for water-splitting hydrogen production.

## 2. Methods

Our first-principles calculations were performed using the projector augmented-wave (PAW) [30] of generalized gradient approximation (GGA) with Perdew–Burke–Ernzerhof (PBE) [31] formalism, implemented in the Vienna ab initio simulation package (VASP) [32,33]. The energy cutoff of the plane-wave expansion was 550 eV. The thickness of the vacuum space was 25 Å in the *z* direction. The reciprocal space was represented by a Monkhorst-Pack scheme with 11 × 11 × 1 *k*-point grid meshes. The convergence criteria of energy for both electrons and ions were, respectively, 1 × 10^−5^ eV and 1 × 10^−6^ eV. Phonon dispersions were retrieved by the density functional perturbation theory (DFPT) as embedded in the PHONOPY code to study the kinetic stability [34,35], for which a 4 × 4 × 1 supercell with 4 × 4 × 1 *k*-point grid meshes was employed. The DFT-D3 van der Waals (vdW)-corrected methods [36] were performed for the improved descriptions of the layered bulk crystals. The calculation results were dealt by the ALKEMIE code [37]. To obtain accurate dielectric functions, we employed the time-dependent Hartree–Fock calculations (TDHF) [38] by the Heyd–Scuseria–Ernzerhof (HSE06) [39] calculations, which have taken the excitonic effects into account.

## 3. Results and Discussion

First, the bulk and monolayer structures of Sc_2_CT_2_ (T = F, Cl, Br) are displayed in Figure 1. The bulk phase is crystallized in the trigonal space group of $P\bar{3}m1$, in which the upper and lower sides of the Sc_2_C plane are halogenated by two T atomic layers. Moreover, the center C atoms are coordinated by six Sc atoms, leading to an octahedron of Sc_6_C. Table 1 lists the calculated lattice parameters *a* (Å) and Sc–C bond lengths *L* (Å) of the Sc_6_C octahedron of the monolayer Sc_2_CT_2_. The lattice parameters *a* (Å) and *c* (Å) of the bulk Sc_2_CT_2_ are listed for comparison as well. According to our calculations, the lattice constant, via the structural optimization for bulk Sc_2_CCl_2_, is 3.445 Å, which is in line with the experimental result [26]. As the surface groups change from F to Br, the lattice parameters increase slightly, which is attributed to the raising of the halogen element atomic radius. In the Sc_6_C octahedron of the monolayer Sc_2_CT_2_, the changes in the Sc–C bond lengths show the same trend as the surface groups from F to Br. It is worth noting that the calculated lattice parameters *a* of the monolayer Sc_2_CT_2_ are almost identical with that of their bulk phases, indicating that the interlayer interaction between the atomic layers are very weak. Thus, the monolayer Sc_2_CT_2_ is likely to be isolated from the corresponding bulk phase.

Herein, the 3D bulk crystals are composed of stacked MXene sheets, as displayed in Figure 1. Distinct from the traditional chemical etching method to fabricate MXene, which is environmentally unfriendly, owing to the using of hydrofluoric acid (HF), we introduce an alternative isolation approach, which has been successfully used for graphene manufacturing. Figure 2 displays the calculated exfoliation energy of Sc_2_CT_2_, and the exfoliation energy of graphite is also shown as a benchmark. As is observed, the evaluated exfoliation energy of graphite is ~0.32 J/m^2^, according to our calculation, which is consistent with the experimental measurement and previous theoretical study [40,41]. Amazedly, the calculated exfoliation energies for all the explored Sc_2_CT_2_ are much lower than that of graphite, suggesting that the monolayer Sc_2_CT_2_ is very likely to be obtained from its bulk phases, by similar experimental methods as graphene. Especially, the exfoliation energy of Sc_2_CF_2_ is only ~0.23 J/m^2^. Such value is nearly two-thirds of that for graphite, indicating the ranking possibility for experimental fabrication.

The stability acts a pivotal part in the practical application and experimental fabrication of 2D Sc_2_CT_2_. Therefore, the phonon dispersion spectra calculations were employed, to assess the kinetic stabilities of 2D Sc_2_CT_2_. Moreover, ab initio molecular dynamics (AIMD) calculations were adopted, to evaluate the thermal stabilities. As the phonon curves illustrated in the upper panel of Figure 3, there is not any appreciable imaginary acoustic dispersion branch of the phonon curve for all the Sc_2_CT_2_, confirming the kinetic stabilities of the three monolayer MXenes. More importantly, the highest phonon spectra frequencies of all the investigated Sc_2_CT_2_ are up to 20 THz. This value is comparable with that of the other monolayer material, implying the rough junction among the Sc, C, and T atoms [19]. The total energy changes and structure snapshots at 300 K, sustaining 9 ps of Sc_2_CT_2_, are displayed in the lower panel of Figure 3. As is shown, the total energies are swinging very slightly and the atoms are well maintained around their equilibrium positions under the temperature field, indicating the thermal stabilities of the monolayer Sc_2_CT_2_. Hence, the kinetic stabilities and thermal stabilities of the Sc_2_CT_2_ monolayer have been approved.

To understand the electronic properties of monolayer Sc_2_CT_2_, the total and partial density of states (DOS) are represented in Figure 4. Note that the DOS of Sc_2_CF_2_, Sc_2_CCl_2_, and Sc_2_CBr_2_ exhibit very similar features, which is in relation to their alike atomic configurations. For the explored Sc_2_CF_2_ and Sc_2_CCl_2_, the CBM is mainly contributed by Sc 3*d* states, while the VBM is entirely dominated by C 2*p* states and Sc 3*d* states. In addition, the C 2*p* orbits and Sc 3*d* orbits are forcefully hybridized [42]. The CBM of Sc_2_CBr_2_ is contributed by Sc 3*d* states as well, but the VBM of it is not only dominated by C 2*p* states and Sc 3*d* states, but also contributed by Br 4*p* states. Moreover, the calculated band gaps for Sc_2_CF_2_, Sc_2_CCl_2_, and Sc_2_CBr_2_ are 1.85 eV, 1.70 eV, and 1.54 eV, respectively, which are endowed with high harvest under the visible-light region. Note that the band gaps of all three 2D Sc_2_CT_2_ are larger than 1.23 eV, which meet the minimum value required for an efficient photocatalyst. Hence, it can be concluded that all the Sc_2_CT_2_ MXenes are promising candidates as semiconductor photocatalysts, in the aspect of bandgaps.

Besides the bandgap size, band edge alignments are another predominant factor to facilitate the water reaction. We have therefore calculated the band edge alignments of Sc_2_CT_2_ in Figure 5, to assess the performance of these newly designed photocatalysts. According to the band alignments in Figure 5, 2D Sc_2_CCl_2_ and Sc_2_CBr_2_ exhibit very similar band edge alignment characters. Interestingly, the VBM of Sc_2_CCl_2_ and Sc_2_CBr_2_ are located at a more positive potential than the water redox potential of H_2_O/O_2_ (1.23 eV vs. normal hydrogen electrode, NHE), implying the remarkable oxidation capabilities of the photo-generated holes. This advantage can be applied in capturing the electrons from organic pollutants, such as methylene blue [19]. However, it is a pity that the CBMs of Sc_2_CT_2_ are also located at a more positive potential than the redox potential of H^+^/H_2_ (0 V vs. NHE), indicating that they are not appropriate for photocatalytic water splitting. While for Sc_2_CF_2_, the CBM is much more negative than the redox potential of H^+^/H_2_ (0 V vs. NHE), but the VBM position is not more positive than the redox potential of O_2_/H_2_O (1.23 eV vs. NHE), suggesting the unapplicable band alignments for water splitting. From the above analysis, one can observe that the band positions of 2D Sc_2_CF_2_, Sc_2_CCl_2_ and Sc_2_CBr_2_ are not suitable as semiconductor photocatalysts under ambition conditions.

From Figure 5, the band alignments of Sc_2_CF_2_ are not appropriate under ambient conditions. However, in the practical application of a photocatalysis, the water redox potential is related to the value of pH in the solutions. Hence, the water oxidation potential of O_2_/H_2_O can be expressed by the following equation [43]:(1)$$E_{O_{2}/H_{2}O}^{OX}=-5.67\;eV+pH\times 0.059\;eV\;$$

According to Equation (1), one can shift the oxidation level of water upward in Figure 5. Thus, the band edge alignments of Sc_2_CF_2_ can be tuned to straddle the redox potential of water. As illustrated in Figure 6, in ambient conditions, Sc_2_CF_2_ is not an appropriate photocatalyst, due to its unsuitable band edge alignments. While in a pH = 8.0 solution, the oxidation level and the reduction potential of water will shift upwards by 0.472 V. We have plotted the band alignments of Sc_2_CF_2_ in Figure 6, from which one can observe that the VBM locates more positively than the hydrogen reduction level (0 V vs. NHE), and the CBM locates more negatively than the oxygen oxidation level (1.23 eV vs. NHE). The results show that Sc_2_CF_2_ has a favorable band position for water splitting in a pH = 8.0 solution. Moreover, the energy difference between VBM and the oxygen reduction potential (oxidizing power) of Sc_2_CF_2_ is 0.25 eV, and the energy difference between CBM and the hydrogen reduction potential (reducing power) is 0.38 eV. Furthermore, we can moderate the water-oxidizing power and reducing power of Sc_2_CF_2_ by changing the pH to 9.0. Notably, the oxidizing power increases to 0.31 eV, while the reducing power decreases to 0.32 eV. Such values suggest that Sc_2_CF_2_ has excellent band edge positions as photocatalysts, which is very helpful for protecting the chemical equilibriums during the water-splitting reaction process.

Having confirmed the band gap and band edge alignments of Sc_2_CF_2_, we now turn to its optical spectra in the visible-light wavelength range, which is another vital criteria for photocatalysts to drive the water-splitting reaction efficiently. Figure 7 displays the optical absorbance of Sc_2_CF_2_, obtained from TDHF-HSE06 and DFT-HSE06, and the light spectrum of silicon is plotted as a benchmark [44]. As is observed, the calculated optical absorbance of 2D Sc_2_CF_2_ are absolutely higher than that of silicon, in most visible-light regions. Moreover, the optical absorption, obtained by the TDHF-HSE06 method, is significantly larger than that evaluated by the standard DFT-HSE06 approach, which implies a strong excitonic effect in Sc_2_CF_2_, in the visible-light region. For the absorption coefficients that were evaluated by the TDHF-HSE06 method, 2D Sc_2_CF_2_ exhibits broad optical absorptions, with three optical spectra peaks at around ~400 nm, ~450 nm, and ~550 nm. It is worth noting that the optical coefficients of Sc_2_CF_2_ can reach up to ~2 × 10^5^ cm^−1^ at around ~550 nm, suggesting an extraordinary light harvesting ability. The outstanding optical performances imply that the Sc_2_CF_2_ monolayer is a very hopeful candidate as a semiconductor photocatalyst for hydrogen generation.

The efficient separation of photo-induced electrons and holes can guarantee high efficiency of a semiconductor photocatalyst. Figure 8a shows the electron localization functions (ELFs) of 2D Sc_2_CF_2_, to investigate the spatial charge distributions. As presented in Figure 8a, the electrons are transferred from Sc atoms to C atoms and F atoms, and completely gathered around C atoms and F atoms, leading to highly localized surroundings. Accordingly, the positively charged region around the Sc atoms will attract the photo-generated electrons, while the negatively charged region around the C and F atoms will trap the photo-activated holes. Hence, the photo-induced electron–hole pairs tend to aggregated in different areas, resulting in efficient carrier separation in the Sc_2_CF_2_ monolayer. As is known, most photo-generated electrons are gathered at the CBM, and photo-generated holes are gathered at the VBM [4,46]. We have then computed the band decomposed charge density of the Sc_2_CF_2_ material in Figure 8. From the band decomposed charge density, we can observe that the CBMs are entirely contributed by the *d* electrons from the Sc atoms, while the VBMs are completely dominated by the Sc atoms and *p* electrons from the C atoms, which is in line with the partial DOS that is displayed in Figure 4a. As is observed, the CBMs are mainly distributed at the Sc–F zones, and the VBM are located at the C zones, thus the photo-induced electrons and holes will transfer to the Sc–F zones and C zones, respectively. Therefore, the photo-excited electron–hole pairs can be separated efficiently at different zones in the monolayer Sc_2_CF_2_, which is a benefit for the photocatalysis water-splitting reaction on the monolayer Sc_2_CF_2_ photocatalyst.

## 4. Conclusions

In conclusion, we have comprehensively explored the photocatalytic water splitting of Sc_2_CT_2_ (T = F, Cl, Br) MXenes, as semiconductor photocatalysts for hydrogen generation, based on first-principles calculations. Firstly, the exfoliation energy of bulk-layered Sc_2_CT_2_ has been calculated, which is much smaller than that of graphite, implying that the monolayer Sc_2_CT_2_ can be isolated by mechanical cleavage from its bulk phases. Additionally, as demonstrated by phonon calculations and ab initio molecular dynamicssimulations, the Sc_2_CT_2_ monolayer shows kinetic and thermal stabilities. Moreover, the calculated band gaps for Sc_2_CF_2_, Sc_2_CCl_2_, and Sc_2_CBr_2_ are 1.85 eV, 1.70 eV, and 1.54 eV, respectively. However, in ambient conditions, Sc_2_CT_2_ is not a suitable photocatalyst, due to its inappropriate band edge position. It is highlighted that the band edge alignments of Sc_2_CF_2_ can be tuned to straddle the redox potential of water, by shifting the oxidation level of water upwards, which can be realized by introducing a pH = 8.0 solution. Furthermore, by changing the pH to 9.0, the oxidizing power and reducing power of Sc_2_CF_2_ can be moderated to a harmonious level, which is very helpful to keep the chemical balance during the redox reaction. Meanwhile, 2D Sc_2_CF_2_ exhibits a high optical absorption, up to 2 × 10^5^ cm^−1^, in visible-light wavelengths. Notably, the photo-excited electron–hole pairs of Sc_2_CF_2_ will separate efficiently. Our exciting results pave the way for designing more MXenes as semiconductor photocatalysts for hydrogen generation.

## Figures and Tables

**Figure 1 materials-14-04739-f001:**
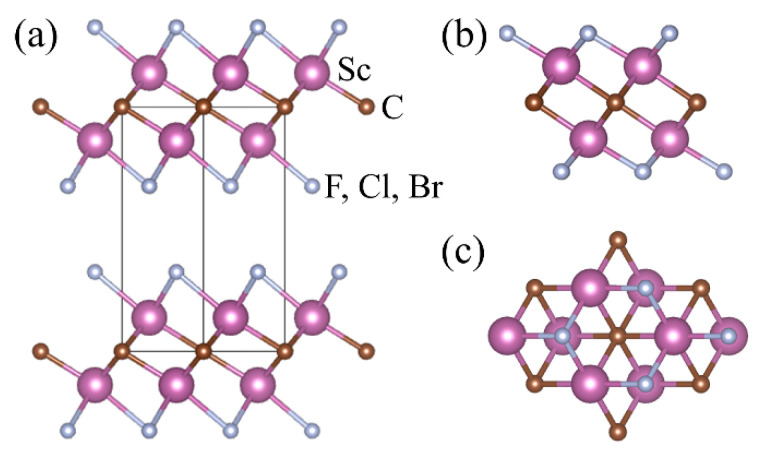
(**a**) Crystal structure of bulk Sc_2_CT_2_ (T = F, Cl, Br). (**b**) Side view and (**c**) top view of single-layer Sc_2_CT_2_.

**Figure 2 materials-14-04739-f002:**
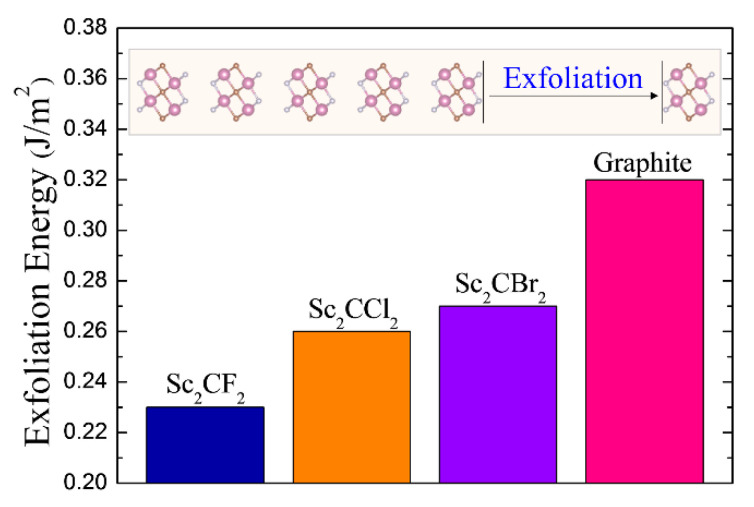
The calculated exfoliation energies of 2D Sc_2_CT_2_ and graphite. The inset is the schematic of exfoliation process.

**Figure 3 materials-14-04739-f003:**
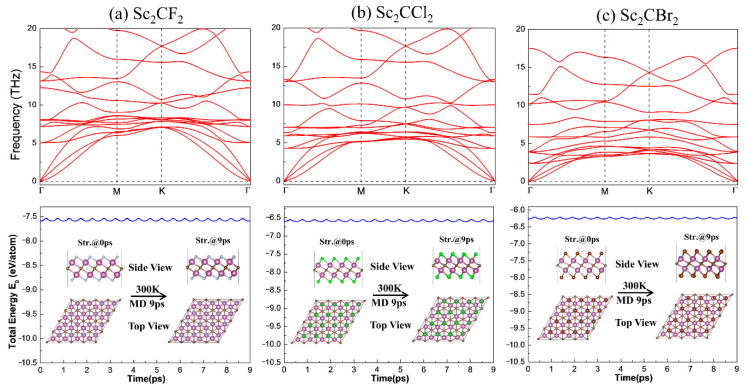
Phonon dispersion curves (upper panel), total energy changes and structure snapshots (lower panel) from 0 to 9 ps from AIMD calculations for (**a**) Sc_2_CF_2_, (**b**) Sc_2_CCl_2_, and (**c**) Sc_2_CBr_2_.

**Figure 4 materials-14-04739-f004:**
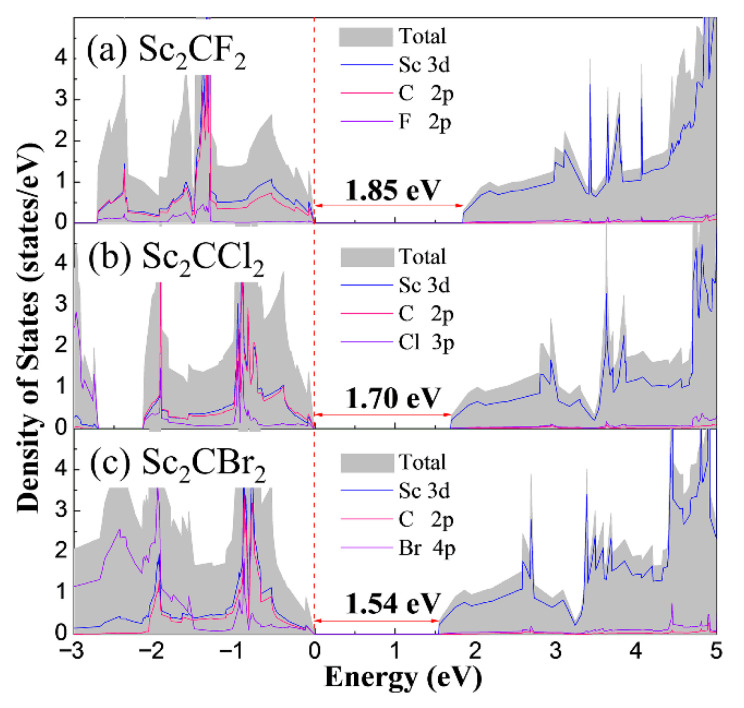
The total and partial density of states of (**a**) Sc_2_CF_2_, (**b**) Sc_2_CCl_2_, and (**c**) Sc_2_CBr_2_. The Fermi energy is marked by dashed lines.

**Figure 5 materials-14-04739-f005:**
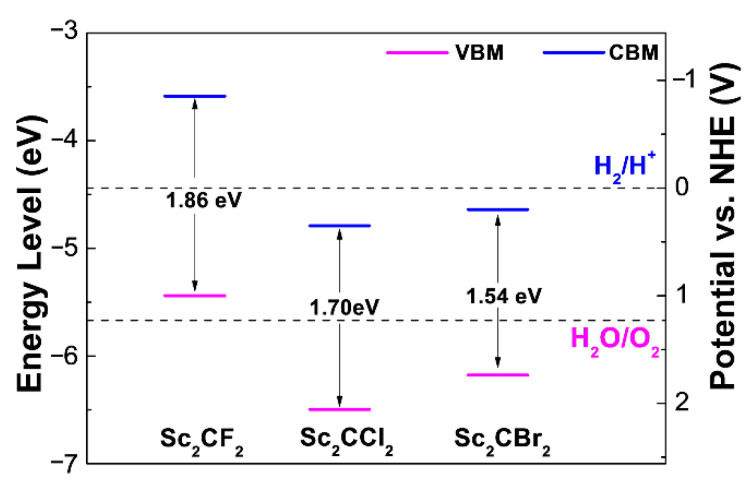
Band alignments with respect to the water reduction and oxidation potential levels of Sc_2_CT_2_ MXenes.

**Figure 6 materials-14-04739-f006:**
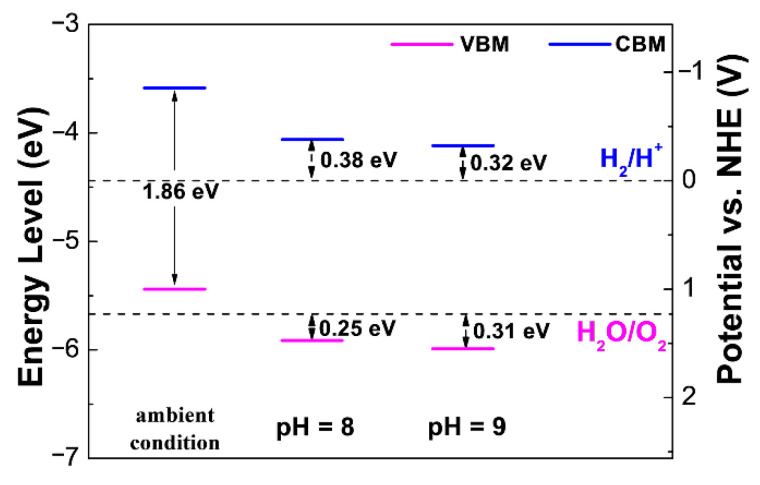
Band alignments with respect to water redox potentials of Sc_2_CF_2_ in ambient conditions, pH = 8 solution, and pH = 9 solution.

**Figure 7 materials-14-04739-f007:**
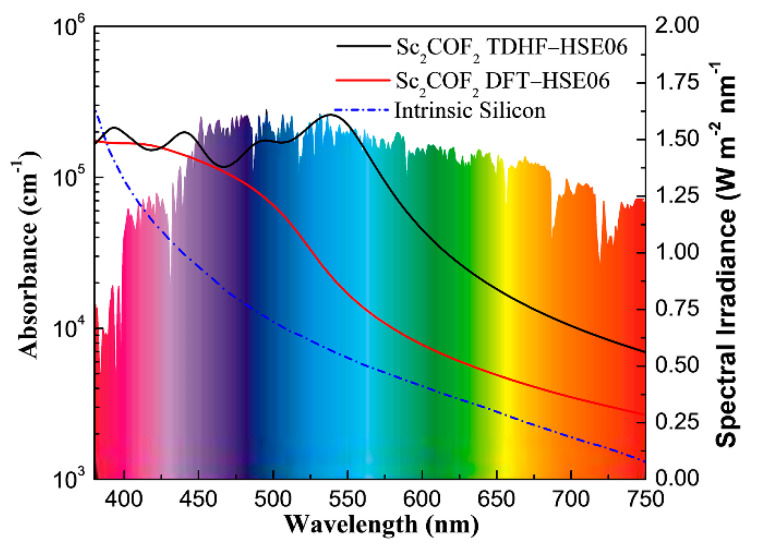
Calculated optical absorption coefficients of Sc_2_CF_2_ compared with the experimental spectrum of intrinsic silicon. The colorful background is the reference solar spectral irradiance [45].

**Figure 8 materials-14-04739-f008:**
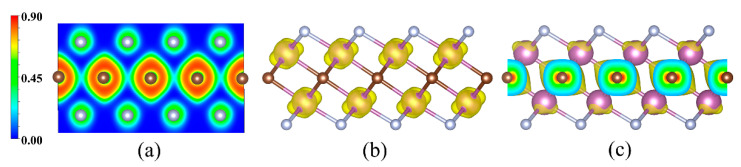
(**a**) The ELF contour plot perpendicular to the (100) direction for 2D Sc_2_CF_2_. The band decomposed charge density of the (**b**) CBM and (**c**) VBM for 2D Sc_2_CF_2_.

**Table 1 materials-14-04739-t001:** The calculated lattice parameters *a* (Å) and *c* (Å) of bulk Sc_2_CT_2_, the lattice parameters *a* (Å) and Sc–C bond lengths *L* (Å) of monolayer Sc_2_CT_2_.

		Sc_2_CF_2_	Sc_2_CCl_2_	Sc_2_CBr_2_
Bulk	*a*	3.285	3.445	3.498
	*c*	6.481	8.854	9.295
Monolayer	*a*	3.252	3.431	3.507
	*L*	2.266	2.330	2.358

## Data Availability

Data available on request. The data presented in this study are available on request from the corresponding author.

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
