# Peer review of "Computational Study of Novel Semiconducting Sc2CT2 (T = F, Cl, Br) MXenes for Visible-Light Photocatalytic Water Splitting"

_materials, 2021, doi:10.3390/ma14164739_

Round 1

Reviewer 1 Report

In this work, the authors theoretically assess the suitability of the Sc2C MXene surface, covered by F, Cl, or Br, for photocatalytic water splitting. The method is similar to the one followed in Ref. 17 (DOI: 10.1039/c6ta04414j), where several MXenes, including Sc2C, covered by F, O, or OH, were studied for the same purpose. In the study of Ref. 17, the Sc2CF2 MXene was deemed unsuitable for photocatalysis of water splitting, because its band alignment was found to be inadequate. In the present study, the same conclusion is reached. However, the authors proceed to consider using different pH conditions in order to shift the band alignment with respect to the water redox potentials, thus achieving a proper environment for photocatalysis. This is a very reasonable approach.

I recommend a major revision for the following reasons. First, the analysis of the Sc2CF2 surface is, for the most part, and up to Figure 5, the same as was done in Ref. 17. Then, after concluding that it is unsuited for photocatalysis, the calculations that follow are, in my view, poorly explained (see the specific comments below). Note that the pH-induced band alignment shift is the truly novel part of the study, and therefore should be given proper emphasis with clearer explanation. Finally, the DOS plots (Figure 4) appear to contradict the text, which also appears to contradict itself in different parts of the manuscript. If the comments presented below are given solid answers, I will gladly recommend this article for publication.

Questions/comments:

1) The manuscript sections and analysis are well organized and easy to follow. There are, however, plenty of minor instances of poor writing, which do not affect the readability or meaning of the text. Nevertheless, because there are so many, and they appear so frequently, they should be corrected. Here are five examples, taken from the Introduction:

Line 34 (verb missing): “the conduction band minimum (CBM) must higher than -4.44 eV”

Line 38 (fragment missing): “To exploit solar radiation more efficiently, an ideal gap 1.8 eV for an efficient photocatalyst.”

Line 39 (verb missing): “Moreover, the efficiency of a photocatalyst always hampered by the recombinations”

Line 40 (maybe “efficient carrier separation”?): “the efficient separation of carrier is another vital point”

Line 42 (pronoun/preposition missing?): “finding an efficient semiconductor photocatalyst for hydrogen generation meet all above requirements is still an ongoing challenge work”

---------------

2) The “Materials and Methods” section could be renamed to just “Methods” because no materials are mentioned in it.

---------------

3) In the plots of Figure 4, the line displaying the highest DOS values in the conduction band is the blue line (Sc 3d), while near the VBM the blue and red lines (Sc 3d and C 2p) are the highest. However, in the text, the authors wrote that “the VBM is entirely dominated by C 2p states and T p states”. In addition, on the Sc2CBr2 VBM, it appears the three lines overlap, so no state is dominating the DOS. Is this a mistake, or did I misinterpret something?

---------------

4) On line 165, the meaning of NHE should be spelled out.

---------------

Equation 1 was taken from Ref. 43. In this Ref., the equation is said to correspond to a chemical equation valid in acidic conditions and includes an extra term due to the partial pressure of O2. I immediately thought of a few questions:

5) Ref. 43 shows two equations for the electrochemical potential, one corresponding to acidic or another for basic conditions. Why did the authors choose the equation for acidic conditions?

6) The authors used Equation 1 to calculate the water oxidation potential at a pH as high as 9. Is this pH still in the “acidic conditions” required to use the equation?

7) Why did the authors exclude the O2 term that was present in the original equation in Ref. 43?

8) MXenes are often extremely oxophilic, and lots of them even adsorb O2 dissociatively, forming a layer of T = O. For this reason, MXenes almost always have oxygen attached. In an experimental environment, is it likely that oxygen may adsorb on, or even replace the F termination of Sc2CF2, and change its catalytic properties?

---------------

9) What is the meaning of the “ambient condition” mentioned several times in the text and in Figure 6?

---------------

10) In Figure 7, the lines for the two HSE06 approaches are of very similar colours. I would suggest changing the colour of one of them (perhaps to black?) to make it easier to distinguish them.

---------------

11) In page 7, the authors mention the partial DOS plots of Figure 4, stating that “CBM are entirely contributed by the hybridization states of d electrons from Sc atoms and p electrons from F atoms, while the VBM are completely dominated by the p electrons from C atoms, which is in line with the partial DOS we have displayed in Figure 4a”. I think this is not exactly in line with the text at the end of page 4, where the authors state that “the CBM is mainly contributed by Sc 3d states, while the VBM is entirely dominated by C 2p states and T p states”. In my opinion, and as I mentioned when commenting Figure 4, this is also not in line with the plots of Figure 4. Could the authors please shed some light on this subject?

Author Response

Response to Reviewer 1 Comments

In this work, the authors theoretically assess the suitability of the Sc2C MXene surface, covered by F, Cl, or Br, for photocatalytic water splitting. The method is similar to the one followed in Ref. 17 (DOI: 10.1039/c6ta04414j), where several MXenes, including Sc2C, covered by F, O, or OH, were studied for the same purpose. In the study of Ref. 17, the Sc2CF2 MXene was deemed unsuitable for photocatalysis of water splitting, because its band alignment was found to be inadequate. In the present study, the same conclusion is reached. However, the authors proceed to consider using different pH conditions in order to shift the band alignment with respect to the water redox potentials, thus achieving a proper environment for photocatalysis. This is a very reasonable approach.

I recommend a major revision for the following reasons. First, the analysis of the Sc2CF2 surface is, for the most part, and up to Figure 5, the same as was done in Ref. 17. Then, after concluding that it is unsuited for photocatalysis, the calculations that follow are, in my view, poorly explained (see the specific comments below). Note that the pH-induced band alignment shift is the truly novel part of the study, and therefore should be given proper emphasis with clearer explanation. Finally, the DOS plots (Figure 4) appear to contradict the text, which also appears to contradict itself in different parts of the manuscript. If the comments presented below are given solid answers, I will gladly recommend this article for publication.

Questions/comments:

1) The manuscript sections and analysis are well organized and easy to follow. There are, however, plenty of minor instances of poor writing, which do not affect the readability or meaning of the text. Nevertheless, because there are so many, and they appear so frequently, they should be corrected. Here are five examples, taken from the Introduction:

Line 34 (verb missing): “the conduction band minimum (CBM) must higher than -4.44 eV”

Line 38 (fragment missing): “To exploit solar radiation more efficiently, an ideal gap 1.8 eV for an efficient photocatalyst.”

Line 39 (verb missing): “Moreover, the efficiency of a photocatalyst always hampered by the recombinations”

Line 40 (maybe “efficient carrier separation”?): “the efficient separation of carrier is another vital point”

Line 42 (pronoun/preposition missing?): “finding an efficient semiconductor photocatalyst for hydrogen generation meet all above requirements is still an ongoing challenge work”

Response 1: We are grateful for the carefully proof and helpful suggestion.

Line 34 “the conduction band minimum (CBM) must higher than -4.44 eV” has been changed to “the conduction band minimum (CBM) must be higher than -4.44 eV”;

Line 38: “To exploit solar radiation more efficiently, an ideal gap 1.8 eV for an efficient photocatalyst.” has been changed to “To exploit solar radiation more efficiently, a value around 1.8 eV is an ideal gap for an efficient photocatalyst.”;

Line 39: “Moreover, the efficiency of a photocatalyst always hampered by the recombinations” has been changed to “Moreover, the efficiency of a photocatalyst always be hampered by the recombinations”;

Line 40: “the efficient separation of carrier is another vital point” has been changed to “the efficient carrier separation is another vital point”;

Line 42 (pronoun/preposition missing?): “finding an efficient semiconductor photocatalyst for hydrogen generation meet all above requirements is still an ongoing challenge work” has been change to “finding an efficient semiconductor photocatalyst for hydrogen generation to meet all above requirements is still an ongoing challenge work”.

Besides the Introduction section, we have corrected the grammatical expressions and improved our English writing for the all manuscript. The corresponding corrections have been highlighted in the revised manuscript.

---------------

2) The “Materials and Methods” section could be renamed to just “Methods” because no materials are mentioned in it.

Response 2: We thank for the important advice. We have renamed the “Materials and Methods” section to “Methods” on line 70.

---------------

3) In the plots of Figure 4, the line displaying the highest DOS values in the conduction band is the blue line (Sc 3d), while near the VBM the blue and red lines (Sc 3d and C 2p) are the highest. However, in the text, the authors wrote that “the VBM is entirely dominated by C 2p states and T p states”. In addition, on the Sc2CBr2 VBM, it appears the three lines overlap, so no state is dominating the DOS. Is this a mistake, or did I misinterpret something?

Response 3: We are grateful for the carefully proof and important issue. We agree the comment of the reviewer that the highest DOS values in the CBM is Sc 3d orbits, and VBM are the C 2p and Sc 3d orbits for Sc2CF2 and Sc2CCl2. Moreover, for the VBM of Sc2CBr2 is not only dominated by C 2p and Sc 3d orbits, but also contributed by Br 4p orbits. The corresponding discussion “For the explored Sc2CF2 and Sc2CCl2, the CBM is mainly contributed by Sc 3d states, while the VBM is entirely dominated by C 2p states and Sc 3d states. In addition, the C 2p orbits and Sc 3d orbits are forcefully hybridized. While for Sc2CBr2, the CBM of which is contributed by Sc 3d states as well, but the VBM is not only dominated by C 2p states and Sc 3d states, but also contributed by Br 4p states.” have been added and highlighted in Page 4 of the revised manuscript.

---------------

4) On line 165, the meaning of NHE should be spelled out.

Response 4: We are grateful for this important issue. The NHE is the abbreviation for Normal Hydrogen Electrode. We have added the explanation on line 166.

---------------

Equation 1 was taken from Ref. 43. In this Ref., the equation is said to correspond to a chemical equation valid in acidic conditions and includes an extra term due to the partial pressure of O2. I immediately thought of a few questions:

5) Ref. 43 shows two equations for the electrochemical potential, one corresponding to acidic or another for basic conditions. Why did the authors choose the equation for acidic conditions?

Response 5: We thank for this rigorous suggestion. Actually, Equation 1 and Equation 2 are dependent and related by the water equilibrium ( ).

6) The authors used Equation 1 to calculate the water oxidation potential at a pH as high as 9. Is this pH still in the “acidic conditions” required to use the equation?

Response 6: In Ref. 43, equation 4 and 5 are mathematically equivalent due to the relation between pH and pOH ( ).

  acidic conditions   (4)

  basic conditions  (5)

And can be simplified to . Herein, the equation can be applied to the pH range from 0 to 14.

7) Why did the authors exclude the O2 term that was present in the original equation in Ref. 43?

Response 7: We consider standard atmospheric pressure in our study, and at 1 atm,  Bar. Hence, the O2 term in the equation is approximately equal to zero, where the equation can be abbreviated as . Therefore, such simplified equation have been taken into consideration in many previous literatures (such as: J. Mater. Chem. C, 2021, 9, 4989-4999; J. Phys. Chem. C 2020, 124, 10385−10397; Chem. Mater., 2013, 25(15), 3232-3238; Phys. Rev. B, 2013, 88(11), 115314). We believe that the estimation of the band edge alignments for the MXenes are reliable.

8) MXenes are often extremely oxophilic, and lots of them even adsorb O2 dissociatively, forming a layer of T = O. For this reason, MXenes almost always have oxygen attached. In an experimental environment, is it likely that oxygen may adsorb on, or even replace the F termination of Sc2CF2, and change its catalytic properties?

Response 8: We thank for the suggestion. In our study, distinct of traditional chemical etching method to fabricate MXene from MAX phase, we have proposed that monolayer Sc2CT2 is likely to be to be isolated from the corresponding bulk Sc2CT2. Thus, the functional group T (T = F, Cl, Br) can be inherited from the bulk Sc2CT2. On the other hand, in our previous work [J. Mater. Chem. A, 2021, 9, 10882], the formation energy of Sc2CO2 and Sc2CF2 has been calculated to evaluate the stability. And the result shows that the formation energy of Sc2CF2 (-12.2 eV) is more negative than that of Sc2CO2 (-9.4 eV), indicating that Sc2CF2 is more energetically favorable than Sc2CO2. Anyway, we agree that the oxygen adsorption process is very important for the photocatalysis water splitting process. However, the reaction dynamic is not the key issue of the current manuscript. Therefore, we would like to work on that in the future.

---------------

9) What is the meaning of the ambient condition mentioned several times in the text and in Figure 6?

Response 9: We thank for the suggestion. The “ambient condition” means the pH = 0, in which the redox potential of O2/H2O is 1.23 eV vs Normal Hydrogen Electrode. Additionally, ambient condition only consider the standard atmospheric pressure at 1 atm for all calculations.

---------------

10) In Figure 7, the lines for the two HSE06 approaches are of very similar colours. I would suggest changing the colour of one of them (perhaps to black?) to make it easier to distinguish them.

Response 10: We thank the reviewer for the carefully proof reading. We have changed the colour of TDHF-HSE06 approach to black and replotted Figure 7 in Page 7 of the revised manuscript.

---------------

11) In page 7, the authors mention the partial DOS plots of Figure 4, stating that “CBM are entirely contributed by the hybridization states of d electrons from Sc atoms and p electrons from F atoms, while the VBM are completely dominated by the p electrons from C atoms, which is in line with the partial DOS we have displayed in Figure 4a”. I think this is not exactly in line with the text at the end of page 4, where the authors state that “the CBM is mainly contributed by Sc 3d states, while the VBM is entirely dominated by C 2p states and T p states”. In my opinion, and as I mentioned when commenting Figure 4, this is also not in line with the plots of Figure 4. Could the authors please shed some light on this subject?

Response 11: We thank for this important suggestion. For Sc2CF2, the CBM are entirely contributed by Sc 3d states, while the VBM are completely dominated by the Sc atoms and electrons from C 2p orbits. We have also revised the statement of Figure 4 (see Response 4). The corresponding discussion “From the band decomposed charge density, we can see that the CBM are entirely contributed by the d electrons from Sc atoms, while the VBM are completely dominated by the Sc atoms and p electrons from C atoms, which is in line with the partial DOS displayed in Figure 4a.” have been added and highlighted in Page 7 of the revised manuscript.

Reviewer 2 Report

In this article, authors studied crystal structures, electronic properties, and optical absorbance of two-dimensional Sc2CT2 (T = F, Cl, Br) MXenes and their corresponding photocatalytic water splitting under visible-light region. To my mind, the article is well written and easy to read.

I think that it can be accepted in the present form.

Author Response

We thank the reviewer for the appreciation of this work.

Reviewer 3 Report

  1. The authors recommend expanding the “Introduction Section” more, considering the study's “Title” in detail.
  2. The study's “Title” should contain the theoretical or DFT term; otherwise, it seems to be entirely experimental work.
  3. The authors should explain the reason behind the lower exfoliation energy for Sc2CF2 over Sc2CCl2 and Sc2CBr2. Importantly, why is it very low comparing graphite?
  4. Why is the bandgap energy for Sc2CBr2 lower than the Sc2CF2. How is it related to their binding energy? What is the role of Fermi level and Energy in deciding those?
  5. The authors recommend demonstrating and discussing at least 2-3 experimental results such as XRD, TEM/HR-TEM, and Photocatalysis in line with their DFT discussions.
  6. English must be improved throughout the manuscript. There are many awkward or grammatically incorrect expressions. Plagiarism should also be checked.

Author Response

Response to Reviewer 3 Comments

  1. The authors recommend expanding the “Introduction Section” more, considering the study's “Title” in detail.

Response 1: We are grateful for the carefully proof and important issue. The corresponding discussion “Thanks to the progress of density functional theory, computational method is one of the most important approach for designing new functional material now.” and “Especially, we have extensively explored the possibility of Sc2CT2 (T = F, Cl, Br) MXenes for visible-light photocatalytic water splitting.” have been added and highlighted in Introduction Section of the revised manuscript.

  1. The study's “Title” should contain the theoretical or DFT term; otherwise, it seems to be entirely experimental work.

Response 2: We thank for the important suggestion. We have revised our study's “Title” as “Computational study of novel semiconducting Sc2CT2 (T = F, Cl, Br) MXenes for visible-light photocatalytic water splitting”.

  1. The authors should explain the reason behind the lower exfoliation energy for Sc2CFover Sc2CCl2and Sc2CBr2. Importantly, why is it very low comparing graphite?

Response 3: We thank for this helpful suggestion. The exfoliation energy is related to the van der Waals force in the layered bulk counterparts. In a previous theoretical work [J. Am. Chem. Soc. 2018, 140, 2417-2420], the exfoliation energy of CrOCl (VOCl) is lower than that of CrOBr (VOBr), and all of them are much smaller than that of graphite. Such phenomenon may be caused by the higher relative atomic mass of Br comparing with Cl atom. Inspiringly, 2D CrOCl has been fabricated in 2019 [ACS Nano 2019, 13, 11353−11362]. That is a very strong proof that DFT study can efficiently design new material.

  1. Why is the bandgap energy for Sc2CBrlower than the Sc2CF2. How is it related to their binding energy? What is the role of Fermi level and Energy in deciding those?

Response 4: We thank for the important advice. We have calculated the electrons transferring for Sc2CT2 (T = F, Cl, Br) to explore this issue. The results show that F in Sc2CF2 take the charge of 0.72e (namely 0.72 electrons transfer from Sc2C to each F atom in average), while the electrons transfer for Br in Sc2CBr2 is 0.58. This phenomenon implies that the bond-bond interaction in Sc2CF2 is stronger than that of Sc2CBr2, which may lead to a higher bonding state energy level and larger bandgap on the basis of the valence-bond theory. For Sc2CT2 (T = F, Cl, Br), the higher bonding energy for C-T bond may make contribute to the larger bandgap according to the electrons transfer calculations. However, as known as we known, the Fermi level is independent with the bandgap value for different MXenes.

  1. The authors recommend demonstrating and discussing at least 2-3 experimental results such as XRD, TEM/HR-TEM, and Photocatalysis in line with their DFT discussions.

Response 5: We thank for this creative advice. In this study, by first principles calculations, we have systemically studied crystal structures, electronic properties, and optical absorbance of two-dimensional Sc2CT2 (T = F, Cl, Br) MXenes and their corresponding photocatalytic water splitting under visible-light region. This is a complete and systemic study, and we will make an effort to fabricate 2D Sc2CT2 experimentally in our future work.

  1. English must be improved throughout the manuscript. There are many awkward or grammatically incorrect expressions. Plagiarism should also be checked.

Response 6: We are grateful for the carefully proof and important issue. We have modified the incorrect grammatical expressions and improved our English writing for the all manuscript. Moreover, we have checked the duplicate according to the journal office and the repetition rate is acceptable for the journal regulation.

Round 2

Reviewer 1 Report

The authors have provided solid answers to all of my questions. The language of the manuscript was improved, as suggested, but grammar issues remain, which, again, do not affect the readibility of the report. Here are a few examples:

Line 58 (Sc2CCl2 is not a device): "Accordingly, the monolayer Sc2CCl2 are promising optoelectronic devices"

Line 65: "Thanks to the progress of density functional theory, computational method is one of the most important approach for designing new functional material now."

Line 98 (verb form/word usage): "As the surface groups change from F to Br, the lattice parameters increases slightly, which is attributed by the raising of halogen element atomic radius"

Line 153 (strange phrasing): "While for Sc2CBr2, the CBM of which is contributed by Sc 3d states as well (...)"

Therefore, my only suggestion is that a minor English language check is performed. Besides this, in my opinion, the manuscript can be published in its current state.

Author Response

The authors have provided solid answers to all of my questions. The language of the manuscript was improved, as suggested, but grammar issues remain, which, again, do not affect the readibility of the report. Here are a few examples:

Line 58 (Sc2CCl2 is not a device): "Accordingly, the monolayer Sc2CCl2 are promising optoelectronic devices"

Line 65: "Thanks to the progress of density functional theory, computational method is one of the most important approach for designing new functional material now."

Line 98 (verb form/word usage): "As the surface groups change from F to Br, the lattice parameters increases slightly, which is attributed by the raising of halogen element atomic radius"

Line 153 (strange phrasing): "While for Sc2CBr2, the CBM of which is contributed by Sc 3d states as well (...)"

Therefore, my only suggestion is that a minor English language check is performed. Besides this, in my opinion, the manuscript can be published in its current state.

Response: We thank the reviewer for the appreciation of the first response letter. We also grateful for the carefully proof.

Line 58 (Sc2CCl2 is not a device): "Accordingly, the monolayer Sc2CCl2 are promising optoelectronic devices" has been changed to “Accordingly, the monolayer Sc2CCl2 are promising optoelectronic materials”.

Line 65: "Thanks to the progress of density functional theory, computational method is one of the most important approach for designing new functional material now." has been changed to “Thanks to the progress of density functional theory, computational method now is a very important approach for designing new functional material.”

Line 98 (verb form/word usage): "As the surface groups change from F to Br, the lattice parameters increases slightly, which is attributed by the raising of halogen element atomic radius" has been changed to "As the surface groups change from F to Br, the lattice parameters increase slightly, which is attributed by the raising of halogen element atomic radius".

Line 153 (strange phrasing): "While for Sc2CBr2, the CBM of which is contributed by Sc 3d states as well (...)" has been changed to “The CBM of Sc2CBr2 is contributed by Sc 3d states as well (...)”.

Reviewer 3 Report

As recommended in "5. The authors recommend demonstrating and discussing at least 2-3 experimental results such as XRD, TEM/HR-TEM, and Photocatalysis in line with their DFT discussions.", the authors should include at least some experimental results.

Author Response

As recommended in "5. The authors recommend demonstrating and discussing at least 2-3 experimental results such as XRD, TEM/HR-TEM, and Photocatalysis in line with their DFT discussions.

Response: We thank for this important advice. For this paper, we have explored two-dimensional Sc2CT2 (T = F, Cl, Br) MXenes by first principles calculations, which are efficient method for design new material. According to our study, the calculated lattice parameter of bulk Sc2CCl2 is 3.445 Å, which is in line with the experimental result (3.435 Å). Moreover, the title has been revised to “Computational study of novel semiconducting Sc2CT2 (T = F, Cl, Br) MXenes for visible-light photocatalytic water splitting”. This is a complete and systemic study, and we will make an effort to fabricate 2D Sc2CT2 experimentally in our future work as soon as possible. While it seems impossible to experimentally study in the revised deadline for 3 days.

This manuscript is a resubmission of an earlier submission. The following is a list of the peer review reports and author responses from that submission.